

# Trials factor for semi-supervised NN classifiers in searches for narrow resonances at the LHC

Benjamin Lieberman[1,2⋆], Salah-Eddine Dahbi[1,3], Andreas Crivellin[4,5],
Finn Stevenson[1,2], Nidhi Tripathi[1,2], Mukesh Kumar[1] and Bruce Mellado[1,2]

**1** School of Physics and Institute for Collider Particle Physics,
University of the Witwatersrand, Johannesburg, South Africa
**2** iThemba LABS, National Research Foundation, Somerset West, South Africa.
**3** Institute of Physics, Academia Sinica, Taipei, Taiwan
**4** Physik-Institut, Universitat Zurich, Winterthurerstrasse 190, CH-8057 Zurich, Switzerland
**5** Paul Scherrer Institut, CH-5232 Villigen PSI, Switzerland

⋆ benjamin.lieberman@cern.ch

## Abstract

To mitigate the model dependencies of searches for new narrow resonances at the Large Hadron Collider (LHC), semi-supervised Neural Networks (NNs) can be used. Unlike fully supervised classifiers these models introduce an additional look-elsewhere effect in the process of optimising thresholds on the response distribution. We perform a frequentist study to quantify this effect, in the form of a trials factor. As an example, we consider simulated $Z\gamma$ data to perform narrow resonance searches using semi-supervised NN classifiers. The results from this analysis provide substantiation that the look-elsewhere effect induced by the semi-supervised NN is under control.

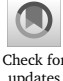

# 1  Introduction

The discovery of the Brout-Englert-Higgs boson [1–3], a scalar resonance, by the ATLAS [4] and CMS [5] collaborations in 2012, marked a significant milestone, completing the particle spectrum of the Standard Model (SM). The endeavour to unearth beyond the SM (BSM) physics hinges significantly on the searches for new particles in the form of narrow resonances. However, the lack of confirmed signals in inclusive or benchmark analyses implies that potential new resonances concealed within the data are likely to be subtle. Such subtlety could be the result of complex production and decay topologies that evade coverage by standard analyses (for a review of the subject see Ref. [6]). Consequently, BSM resonances might have evaded detection, lurking unnoticed within uncharted or inadequately explored regions of the phase space.

    Machine Learning has the substantial promise of addressing this challenge. It has already demonstrated considerable effectiveness in probing areas of special interest, exemplified by the detection of the SM Higgs boson. These "standard" fully supervised techniques, which rely on the knowledge of event-by-event truth-level information, called "labels", can falter if the manifestation of BSM physics diverges from the model used as a reference. This means that if the simulated data on which these models are trained is too specific or flawed, due to bias or insufficient truth-level information, the model will learn these artifacts of the simulation and might not recognise inherent signals. To address this challenge, model-independent searches utilise unsupervised [7–11], semi-supervised [12,13] or weakly supervised [14–16] techniques. Unsupervised techniques do not rely on any labels for events, thus avoiding any dependency on a model. In contrast, semi-supervised techniques apply labels only to background events, enabling a model-independent extraction of signal events. Weakly supervised techniques, on the other hand, utilize the proportions of signal to background events in the training samples, limiting the model dependence by relying on partial labeling. These techniques are therefore most often used to look for deviations from the background-only hypothesis.

    The classification without labels (CWoLa) study, presented by the ATLAS collaboration, introduces weak learning techniques to reduce the dependency of the recognition mechanism on simulated data [17]. The study uses mixed samples (combining signal and background events in each sample) to train binary classification models. It reveals that weakly supervised models can classify signal events with success comparable to the fully supervised method [18–21]. Variations of the weakly supervised method have been successfully implemented for resonance detection in Refs. [22–33]. An advantage of the model agnostic, weakly and semi- supervised, methods is the ability to be trained on impure[1] mixed samples [34], or directly on data, reducing the need for simulation. Research on the use in particle physics has been extended to include multi-model classifiers [35], graph neural networks [36], with the potential inclusion

---

[1]An impure sample refers to a dataset containing not only the signal and background of interest, but also additional signals.

of transfer learning [37]. Finally, the viability of the model-independent methodologies for narrow resonance searches was studied in Refs. [23, 26] and a review of weak supervision was presented in Ref. [38].

This weakly-supervised method can be adapted to use a labeled background sample containing no signal events, alongside an unlabeled mixed sample containing both signal and background events. This methodology allows the classifier to learn to distinguish background events from signal events by being trained on a background-only sample of simulated events. By subtracting the learned background events from the mixed sample, one is left only with the signal events of interest. The mixed sample can therefore consist of actual LHC data, allowing for a model-independent extraction of signal events. This semi-supervised technique, implemented via a side-band analysis, is therefore an extension of the weak supervision method ideal for reducing model dependency and detect BSM physics with topological variations between the possible signal and side-band regions [39].

When conducting any resonance search, it is crucial to take into account the look-elsewhere effect to arrive at a statistically meaningful result, i.e. significance for an excess. This effect refers to the increased probability of finding apparent signals (that are actually statistical fluctuations) when searching over an extended parameter range, usually mass. When using semi-supervised classifiers, an additional look-elsewhere effect is introduced. This is due to the fact that the signal sample, used for training, is not predefined like in the fully supervised method and a scan must be performed on the response distribution to determine the optimal threshold cut. This scan of the response distribution introduces an additional look-elsewhere effect that can manifest itself as "fake signals" which can significantly distort resonance patterns and influence their interpretations [40]. Hence, it is imperative to evaluate the probability of fake signals produced in the implementation of semi-supervised classifiers and expose it in the form of a trials factor. Importantly, this trials factor has to be included on top of the trials factor taking into account the probability of statistical fluctuations over e.g. an extended mass range, and thus occurs even if the mass of the new resonance is fixed (known).

In general, model-independent searches for narrow resonances are of great interest. In recent years, the field of particle physics has experienced a growing plethora of anomalous experimental results (for recent reviews, see Refs. [41, 42]). Many of them are statistically significant, continue to grow, and remain unexplained by state-of-the-art SM calculations. While some anomalies might eventually find resolution within the SM, persisting ones could contain signs of new physics and may serve as a guide to model building and experimental searches. Of particular relevance to this paper are the multi-lepton anomalies at the LHC [43–46]. These can be explained by introducing (at least) two additional scalar fields, $H$ and $S$ [43, 47], where, e.g. $H \to SS^{(*)}$. The scalar $H$ has a mass in the ballpark of 270 GeV, while the mass of $S$ was determined to be $m_S = 150 \pm 5$ GeV [44]. The first indication of a narrow resonance within this mass range of the di-photon spectrum was reported at $m_S = 151.5$ GeV in Ref. [48] and further corroborated with more data in Ref. [49].

This provides a compelling reason to seek Higgs-like resonances near the electroweak (EW) scale, an area where SM backgrounds are quite significant. The use of topological constraints through semi-supervision could be crucial in this context, as they help to dampen the impact of these backgrounds, while reducing model dependencies in the search. The $Z\gamma$ final state dataset is therefore selected as an excellent illustrative example for the proposed methodology.

In this paper, we focus on the $S(151.5)$ candidate which is expected to be produced in associated production, i.e. jointly with a litany of different final states, and consider a simulated $Z\gamma$ dataset. For this we present the methodology and result of a frequentist approach to estimate the look-elsewhere effect introduced in the training and implementation of semi-supervised neural network (NN) classifiers. The frequentist approach involves repeating a pseudo-experiment many thousands of times, before using the distribution of results to quan-

tify the probability of observed outcomes. Motivated by the multi-lepton anomalies and the excess around 150 GeV, each pseudo-experiment is designed to expose the significance of fake signal, found around this mass, after training the model and extracting events from its response. Additionally, in order to obtain a sufficient quantity of simulated training data, to implement the frequentist study, an evaluation of ML-based data generators for scaling and sampling HEP datasets is presented.

## 2  Simulated dataset

Our objective is to investigate the likelihood that a semi-supervised NN produces a false signal. This involves determining the probability that, in the absence of any actual signal events, the NN still identifies or creates resonance signatures. Therefore, we only need to consider and simulate the background events for the corresponding search. However, since we need many pseudo-experiments for a frequentist study, we will not only rely on Monte Carlo (MC) methods for generating this background but also use machine learning tools for scaling the MC events in a computationally efficient way.

### 2.1  Monte Carlo simulation

The dominant background for our $S(150\,\text{GeV}) \rightarrow Z\gamma$ search is the production of a prompt photon in association with a $Z$ boson [50], comprising over 92% of the total background yield [51]. Our simulation accounts for the fact that the $\ell^+\ell^-\gamma$ invariant mass of 130 GeV to 170 GeV necessitates the $Z$ boson to be off-shell. We configured `MadGraph5 aMC@NLO 2.6.7` to include the off-shell $Z$ boson and virtual photon mediated diagram ensuring a comprehensive representation of the physics involved.

The event generation utilized next-to-leading-order (NLO) parton distribution functions, `NNPDF 3.0`, combined with NLO matrix elements to ensure consistency and accuracy [52]. Generator-level cuts were applied during the event generation in `MadGraph5 aMC@NLO 2.6.7`, specifically imposing an invariant mass cut on the $\ell\ell\gamma$ system to select events within the 130-170 GeV range. These generated events were then processed through `PYTHIA 8.2` [53] for parton showering and `Delphes 3` [54] for detector simulation, where reconstructed-level cuts were applied to mimic experimental conditions. Reconstructed-level cuts included detailed pre-selection criteria such as the number of leptons and photons, their transverse momenta, and isolation requirements. Specifically, the $\ell^+\ell^-\gamma$ candidates were selected by requiring exactly two oppositely charged same-flavor leptons with an invariant mass greater than 40 GeV and one isolated photon with $E_T^\gamma > 25\,\text{GeV}$.

Jets were reconstructed using the anti-$k_t$ algorithm [55] with the radius parameter $R = 0.4$, as implemented in the `FastJet` 3.2.2 [56] package, and jets with $p_\text{T} > 30\,\text{GeV}$ and $|\eta| < 4.7$ are considered. Reconstructed jets overlapping with photons, electrons, or muons in a cone of size $R = 0.4$ are removed as well as photons that were within a $\Delta R < 0.4$ of any selected lepton. Electrons and muons are required to have $p_\text{T} > 25\,\text{GeV}$ and $|\eta| < 2.5$. The central jets are those with a pseudorapidity within the range $|\eta| < 2.5$. This cut is applied to ensure that the jets fall within the region of the detector where the tracking and calorimeter systems have optimal performance. Overlap removal procedures were implemented to avoid double-counting objects.

A sample of 110′000 SM $Z\gamma$ MC events are simulated for this analysis. The observables to be used as kinematic features in our study are:

- The $Z\gamma$ invariant mass ($m_{\ell\ell\gamma}$).

- Missing transverse energy ($E_T^\text{miss}$).

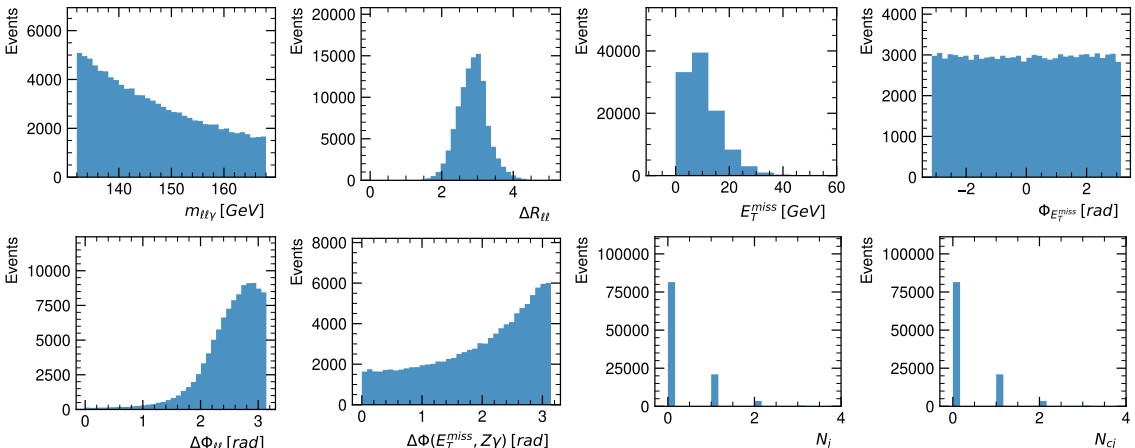

Figure 1: Selected kinematic distributions of the MC generated $Z\gamma$ background dataset consisting of 110′000 events.

- Missing transverse energy azimuthal angle ($\Phi_{E_T^{\mathrm{miss}}}$).

- The distance between the two leptons in the $\eta - \phi$ space ($\Delta R_{\ell\ell}$[2]).

- The difference between the azimuthal angle of the two leptons ($\Delta\Phi_{\ell\ell}$).

- The difference between the azimuthal angle of the missing transverse energy and $Z\gamma$ system ($\Delta\Phi(E_T^{\mathrm{miss}}, Z\gamma)$).

- The number of jets ($N_j$), and the number of central jets ($N_{cj}$).

The feature distributions of the MC dataset are shown in Figure 1. All observables, excluding $m_{\ell\ell\gamma}$, are used as inputs for the NN classifier. The $m_{\ell\ell\gamma}$ distribution, and any features correlated with it, cannot be used to train the NN due to the fact that the training samples, mass window ($144 < m_{\ell\ell\gamma} < 156\,\mathrm{GeV}$) and side-band ($132 < m_{\ell\ell\gamma} < 168\,\mathrm{GeV}$ excluding mass window), are defined on this mass.

## 2.2 Data sampling and generation

Frequentist studies require a sufficient number of repetitions of an experiment to have statistically meaningful results. For each iteration of the pseudo-experiment, a sample of 80′000 events is used to train and evaluate the NN classifier, in order for the results not to be limited by the MC statistics. However, repeatedly generating these samples of events with pure MC methods would require substantial computational resources and time [57]. Fortunately, machine learning-based data generators have recently emerged as a novel and powerful tool for generating and scaling data samples [58, 59]. These advanced algorithms leverage the potential of ML and deep learning to create realistic, high-quality simulated datasets. Such data generators utilize various architectures, most notably Generative Adversarial Networks (GANs) and Variational Autoencoders (VAEs), each demonstrating unique capabilities and potential use-cases in particle physics.

To generate a sufficient quantity of high-quality data for this frequentist analysis, the three best-performing methods were found to be the Wasserstein GAN (WGAN), the VAE with an additional discriminator (VAE+D), and the Kernel Density Estimation (KDE). The model architectures, training mechanisms and resultant plots, for each method, are presented in Appendix B.

---

[2]The distance $\Delta R$ between two leptons in the $\eta - \phi$ space is defined as $\Delta R = \sqrt{(\Delta\eta_{\ell\ell})^2 + (\Delta\phi_{\ell\ell})^2}$.

Table 1: Comparative evaluation of the WGAN, VAE+D and KDE data generation techniques. This table presents the mean and standard deviation of evaluation metric results for each model, based on 500 generated $Z\gamma$ samples, to assess their performance.

| Evaluation Metric | WGAN | VAE+D | KDE |
|---|---|---|---|
| Relative difference | $2.67 \times 10^{-3}$ $(\pm 3.1 \times 10^{-5})$ | $4.42 \times 10^{-3}$ $(\pm 2.2 \times 10^{-5})$ | $3.21 \times 10^{-4}$ $(\pm 1.6 \times 10^{-5})$ |
| Kolmogorov-Smirnov score | $2.67 \times 10^{-2}$ $(\pm 2.89 \times 10^{-4})$ | $5.55 \times 10^{-2}$ $(\pm 3.04 \times 10^{-4})$ | $4.50 \times 10^{-2}$ $(\pm 2.18 \times 10^{-4})$ |
| Spearman correlation coefficient | $6.96 \times 10^{-1}$ $(\pm 5.99 \times 10^{-3})$ | $6.23 \times 10^{-1}$ $(\pm 7.07 \times 10^{-3})$ | $9.38 \times 10^{-1}$ $(\pm 1.37 \times 10^{-2})$ |
| Spearman p-value | $3.43 \times 10^{-23}$ $(\pm 5.19 \times 10^{-23})$ | $1.38 \times 10^{-17}$ $(\pm 1.67 \times 10^{-17})$ | $8.95 \times 10^{-54}$ $(\pm 1.84 \times 10^{-52})$ |
| Correlation difference | $2.14 \times 10^{-2}$ $(\pm 2.84 \times 10^{-4})$ | $2.62 \times 10^{-2}$ $(\pm 3.27 \times 10^{-4})$ | $2.47 \times 10^{-3}$ $(\pm 2.11 \times 10^{-4})$ |

The models are each trained and evaluated on the previously generated $Z\gamma$ MC dataset. The quality of the data produced using each method is evaluated with the following metrics to compare them to the MC training data. This is firstly implemented in terms of the feature distributions, using the bin-wise relative difference and Kolmogorov-Smirnov score [60]. Furthermore, the events feature-wise correlation is evaluated using the Spearman correlation coefficient [61] and the absolute difference between the training and generated correlations, to expose the extent to which the generated events reflect true physics. To evaluate the model's ability to produce many samples of sufficient quality, the mean evaluation metric results for 500 generated samples (of 80′000 events each), using the different methods, are summarised in Table 1.

The result of comparing the different data generation methods demonstrates that the WGAN, VAE+D and KDE are each able to simulate physics events of excellent quality. The best-performing model to generate data for the frequentist study is shown to be the KDE method. An example set of pseudo-experiment data generated by the KDE is presented in the form of feature distributions, in Figure 2, and feature correlations, in Figure 3.

# 3 Methodology

Machine learning classifiers use the underlying patterns within the data to obtain a designated output. In particle physics, this process has the potential to introduce fake signals. This means that using a NN to observe an excess might be more probable than what is expected based on statistical fluctuations only. This can influence the statistical significance of any discovery of a narrow resonance. To evaluate the propensity of semi-supervised NN classifiers to introduce these fake signals, a frequentist methodology is used here.

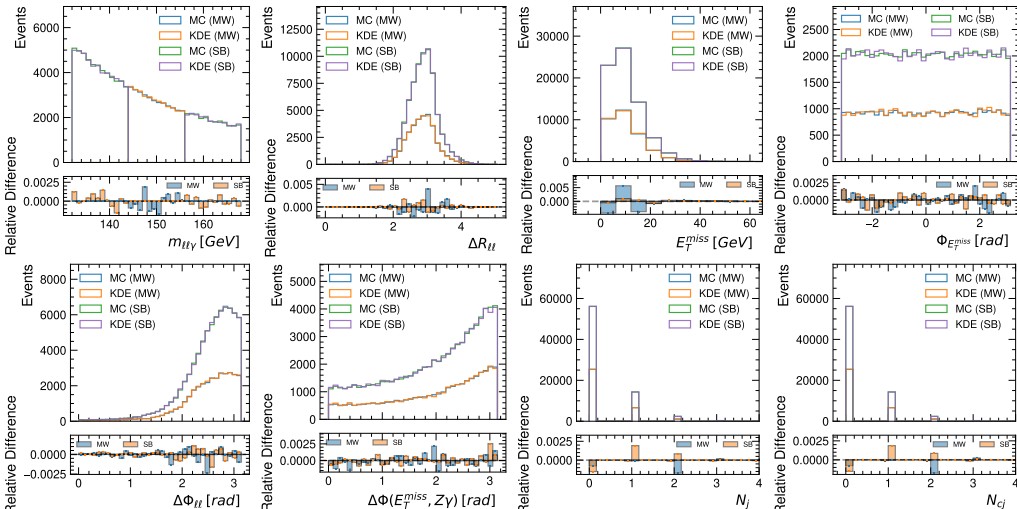

Figure 2: Feature distribution comparison between the MC and KDE generated events, for one example of the pseudo-experiments. Each feature distribution is divided into events in the mass-window (MW) and side-band (SB).

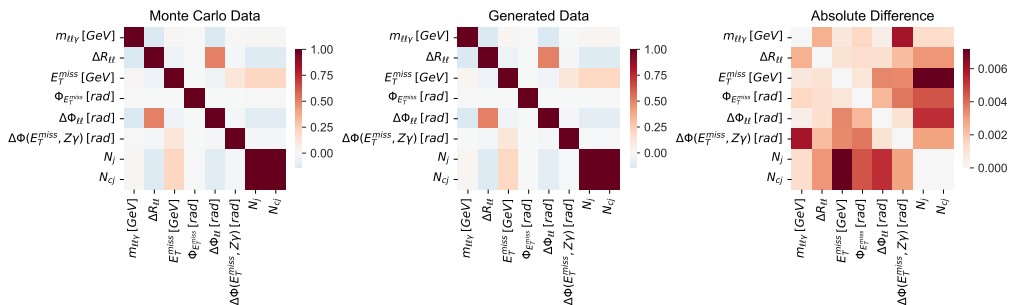

Figure 3: Feature correlation comparison between the MC and KDE generated events, for an example pseudo-experiment. Left: feature correlation of MC training dataset. Middle: feature correlation of KDE generated dataset. Right: the absolute difference between the feature correlations of the MC and KDE generated datasets.

A frequentist study evaluates probabilities by repeating a pseudo-experiment many times. The pseudo-experiment in this study consists of training and evaluating a semi-supervised NN classifier on samples of SM $Z\gamma$ KDE generated events. The steps of the pseudo-experiment, shown in Figure 4, include data sampling, NN training, background rejection scans and the calculation of the local signal significance for an excess in the signal region. Each of these components are described in detail in the following sections.

## 3.1 Semi-supervised neural network

The Neural Network classifier used in this study is implemented using the PyTorch framework [62]. The model architecture consists of an input layer, three hidden layers and an output layer. The input layer has seven neurons, each corresponding to one of the selected input features, described in Section 2. The number of neurons in each of the hidden layers are 26, 26 and 13, respectively. The hidden layers each use the hyperbolic tangent activation function, *Tanh* [63]. The output layer consists of one neuron and is activated by a *Sigmoid* function, constraining the output response to a value between zero and one.

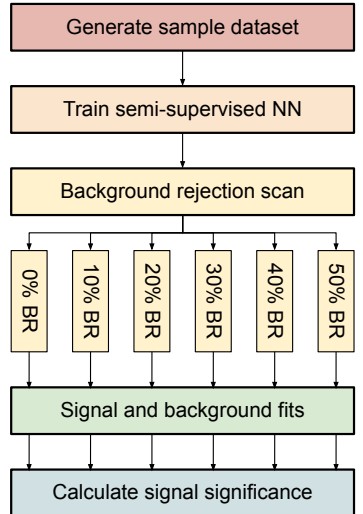

Figure 4: Diagram illustrating the methodology for a pseudo-experiment to assess possible trails factors when using a semi-supervised NN in the search for a narrow resonance. The process starts with the generation of a sample dataset, followed by the training of the semi-supervised NN. Subsequent stages include a background rejection (BR) scan to extract local BR samples from the NN response, fitting signal and background models to each local sample, and finally the computation of the signal significance.

For each iteration of the pseudo-experiment, the NN model is trained independently, on the corresponding data sample, for 50 epochs using the Adam [64] optimiser, a learning rate of $10^{-3}$, and a batch size of 256 events. The training uses a background sample and a mixed sample. The background sample consists of the events in the side-band regions which are labeled as background (0). The mixed sample is made up of events within the mass window of interest, and are labeled as signal (1). As only $Z\gamma$ data intended as a background is used, no actual signal events are present in the mixed sample, therefore the training samples should be indistinguishable to the NN apart from statistical fluctuations. This means that the NN should not be able to discriminate between the samples to the extent that any excess of signal events found in the signal region can be considered a fake signal.

## 3.2 Background rejection scan

Once trained, the NN model is used to infer a response value for each event, where values tending to zero indicate background and events tending to one indicate signal. In the semi-supervised context, the extraction of signal events using the NN response can be understood as the rejection of background events. By rejecting background events, which are events with a response tending to zero, a sample of signal events is left. The optimal background rejection cut on the response distribution must therefore be determined to remove as many background events as possible from the sample. To simulate the process of optimising the background rejection cut, to maximise the purity of the signal sample, a scan of cuts to the model's response distribution is implemented. When conducting a statistical analysis in which a scan is performed, the extent of the corresponding look-elsewhere effects must be considered.

To understand this additional look-elsewhere effect, the background rejection scan is used to extract samples of events with increasing amounts of rejected background events. The amount of background events rejected in each sample is selected to be 0, 10, 20, 30, 40 and

50 percent of the total events respectively. As there are no signal events present to classify, the defined cuts provide samples of events with increasing influence of background rejection, while maintaining at least half of the event statistics. The six batches of events extracted, for each background rejection, can therefore each be considered a local analysis sample. For a given pseudo-experiment, each local sample will be fitted to measure a local resultant signal significance, revealing any fake signals in the sample. To this end, each local sample is mapped to its respective invariant mass for further analysis.

### 3.3 Significance calculation for a resonance in the invariant mass distributions

In this analysis, the resonant anomaly detection search strategy is used to discover evidence for BSM physics. The new physics signal is expected to originate from a decay of a new particle, $S$, in the form of a resonance in the corresponding invariant mass. The investigation therefore involves searching for an excess with respect to the continuous SM background. Since we want to examine the probability of a fake signal, we define a signal region (or mass window) in the invariant mass, $m_{\ell\ell\gamma}$, where an excess of signal events can be observed. We select a fixed center of the signal mass window of $m_{\ell\ell\gamma} \approx 150\,\text{GeV}$, motivated by the corresponding excess [44, 46, 48, 49]. The background (or side-band) region is defined as the mass range surrounding the signal region where only SM background events are expected. The signal (mass-window) range is defined as $144 \leq m_{\ell\ell\gamma} \leq 156\,\text{GeV}$ and the background (side-band) range as $132 < m_{\ell\ell\gamma} < 168\,\text{GeV}$ excluding the mass-window.

The invariant mass distribution of each background rejection sample is used to calculate the significance of a local excess, if present, in the prescribed mass window. To determine the extent of any signal excesses, a background-only hypothesis is used. The background is defined using the functional form expressed in Equation 1 in Ref. [48]:

$$f(\epsilon_m) = (1 - \epsilon_m)^{c_0} \cdot \epsilon_m^{c_1 + c_2 \cdot \log(\epsilon_m)}, \tag{1}$$

where $c_0$, $c_1$ and $c_2$ are fit parameters and $\epsilon_m$ is the invariant mass, $m_{\ell\ell\gamma}$, divided by the center of the signal mass window energy, i.e. 13 TeV. Excesses of signal events within the mass window are captured using a Gaussian function, Equation 2:

$$g(m_{\ell\ell\gamma}) = c_s \cdot e^{-\frac{(m_{\ell\ell\gamma} - \mu_s)^2}{2 \cdot \sigma_s^2}}, \tag{2}$$

where $c_s$ is the parameter measuring the height of the peak, $\mu_s$ is the position of the center of the peak, and $\sigma_s$ is the search resolution. In this analysis, the $\mu_s$ is fixed at 150 GeV and $\sigma_s$ is set to the search resolution of 2.4 GeV [51].

The probability density function, composed of the signal and background functions, is minimised using the negative log-likelihood, allowing any signal events to be captured by the Gaussian function. By integrating over the mass range for the signal and background fits, the number of signal and background events can be extracted. Using the number of signal and background events, the local significance, $Z$, can be calculated as

$$Z = \sqrt{2 \cdot \left( (N_s + N_b) \cdot \log\left( 1 + \frac{N_s}{N_b} \right) - N_s \right)}, \tag{3}$$

where $N_s$ and $N_b$ are the number of signal and background events respectively.[3] Each pseudo-experiment will therefore produce six local significance values, reflecting each background rejection sample. An example of the signal and background fits for a single iteration of the pseudo-experiment are shown in Appendix A, Figure 7.

---

[3]Note that the systematics related to the background fitting functions, i.e. the spurious signal analysis, are not included here. However, this uncertainty should be included on top of the additional look-elsewhere effect studied here and is not related to the use of NNs.

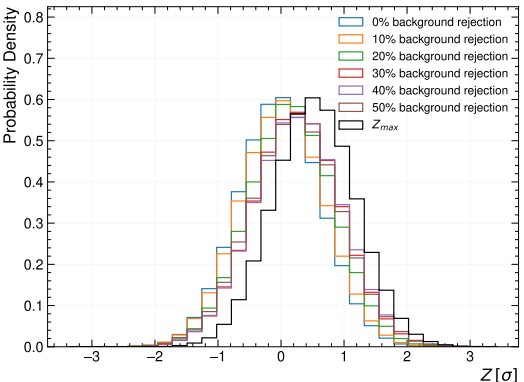

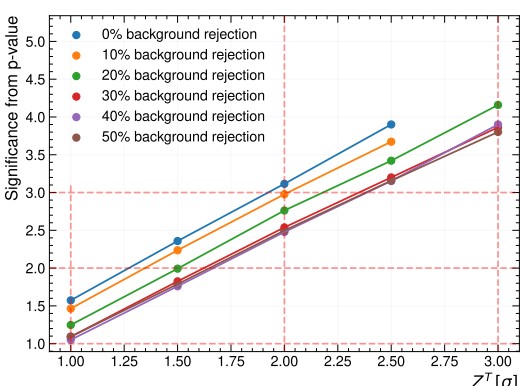

(a) PDFs of the significance for an excess at 150 GeV obtained from the pseudo-experiments for each background rejection category, $f(Z)_{\mathrm{BR}}$, and for the maximum significances, $f(Z)_{\max}$.

(b) Positive background rejection p-values (converted to significance) as compared to significance thresholds.

Figure 5: PDFs and comparison to a standard Gaussian for the local samples, i.e. the different background rejection cuts.

## 4 Results and discussion

We use a frequentist approach to explore the size of the additional look-elsewhere effect originating from the implementation of the semi-supervised classifier. To this end, the pseudo-experiment presented in Section 3 is repeated $125'000$ times to obtain the necessary statistics. This means that for each pseudo-experiment the KDE model, pre-trained on the $Z\gamma$ MC-generated dataset of $110'000$ events, is used to generate a new sample of $80'000$ statistically independent events. These events are then used to train the semi-supervised NN. Then, the event samples for the background rejection cuts of 0%, 10%, 20%, 30%, 40% and 50% are extracted via the NN response distribution. The invariant $Z\gamma$ mass distributions of these samples are then used to calculate the significance for the existence of a narrow resonance at 150 GeV. Therefore, each pseudo-experiment will produce six significance values, corresponding to each of the background rejection thresholds.

This methodology, similar to mass scans in high-energy physics, segments our analysis through the use of background rejection (BR) samples, where each threshold acts as a category. Each category, however, is not statistically independent as the events in each sample overlap. This approach facilitates an examination of the impact of background rejection levels on our results, enabling a nuanced understanding of their influence on our findings.

The probability density functions (PDF) of the significances for the six different BR cuts ($f(Z)_{\mathrm{BR}}$) as well as the distribution of maximum significances, $f(Z)_{\max}$, achieved per pseudo-experiment across all BR cuts are shown in Figure 5a. The $f(Z)_{\mathrm{BR}}$ distributions have an approximately Gaussian shape, however with a standard deviation between 0.65 and 0.71. This decrease in variance w.r.t. the standard Gaussian is due to the constraints within the resonance fit imposed a priori via the shape of the resonance and the side-bands definitions. Note that the peak of the PDFs shifts to the right with an increasing background rejection cut. Such a shift emerges from the interplay between signal and background modelling, amplified by the diminishing event counts in samples subjected to higher background rejection cuts.

To further interpret these results and quantify the deviations from a Gaussian shape, the probability of getting a significance of up to a significance threshold, $Z_T$, of $1\sigma$, $2\sigma$ and $3\sigma$ can be calculated for each BR cut sample. For this, we consider only positive values of the significances for an excess at 150 GeV, as only they can reflect instances where a fake signal

is generated. The corresponding normalised probability density function is

$$f^+(Z)_{\text{BR}} = \frac{f(Z)_{\text{BR}}}{\int_0^\infty f(Z)\,dZ}\,, \qquad \text{for } Z > 0\,, \tag{4}$$

where BR is a label indicating the background rejection cut. We then compute the probability that, for each background reject cut, a significance of up-to a value $Z^T = 1\sigma$, $2\sigma$ and $3\sigma$ is obtained via the cumulative distribution function (CDF):

$$F(Z^T)_{\text{BR}} = \int_0^{Z^T} f^+(Z)_{\text{BR}}\,dZ\,. \tag{5}$$

Therefore, the corresponding $p$-values calculated as $1 - F(Z^T)_{\text{BR}}$, quantify the probability of detecting an excess of up-to $Z^T$. These $p$-values can be converted to significance, before being directly compared to $Z^T$, corresponding to if the distribution were normally distributed. This relationship is shown in Figure 5b, where one can see that the BR samples conform to a large degree with expected probabilities from a Gaussian distribution. Only at higher significance thresholds above $\approx 2.5\sigma$ slight deviations from the diagonal are seen. It is also shown that for background rejection cuts of 0% and 10%, no $3\sigma$ significance excesses were found during the $125'000$ iterations. Note that, in Figure 5b, categories with lower background rejection differ more greatly from the nominal values due to decreased positive significance yields across all thresholds.

An insight into the influence introduced by the semi-supervised classifier, is to consider the 0% BR sample as a baseline. This inclusive sample contains all the events classified by the NN without any cuts applied, and therefore provides a sample unaffected by the influence of the classifier and its response scan. An interpretation of additional effects can thus be revealed through a comparison of the samples that include the influence of the NN, from response cuts, to the inclusive baseline. The comparison of the $p$-values from the different BR cut samples to the 0% BR cut as a function of the significance threshold $Z^T$, are shown in Figure 6a.

To quantify the trials factor introduced by the semi-supervised classifier, we compare the probability of obtaining the maximum significance values across all BR categories and pseudo-experiments to the expected nominal probabilities for each significance threshold.

The "global" distribution is constructed as the set of maximum significance values, $Z_{\text{max}}$, obtained from each pseudo-experiment by considering all background rejection cuts. The corresponding global p-value for a given significance threshold, $Z^T$, is calculated as:

$$p_{\text{global}}(Z^T) = 1 - \int_0^{+Z^T} f^+(Z')_{\text{max}}\,dZ'\,, \tag{6}$$

where $f^+$ is defined analogously to Eq. (4).

To assess the inflation of significance due to the look-elsewhere effect, we compute the trials factor, which is the ratio of the global p-value derived from the $Z_{\text{max}}$ distribution to the local (nominal) p-value for each significance threshold $Z^T$. This trials factor represents the inflation in significance due to the look-elsewhere effect introduced by considering multiple BR categories. The trials factor is defined as:

$$\text{Trials Factor} = \frac{p_{\text{global}}(Z^T)}{p_{\text{local}}(Z^T)}\,. \tag{7}$$

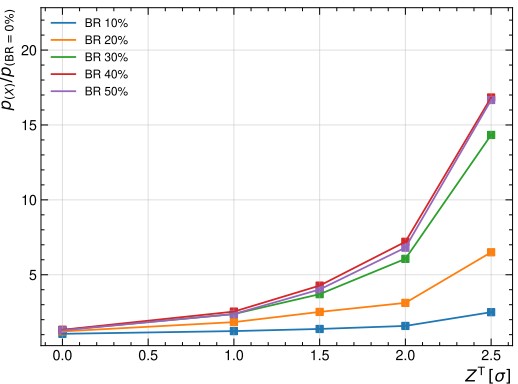

(a) The ratio $p_{(X)}/p_{(BR=0\%)}$ for $X = 10\%$, 20%, 30%, 40%, 50% BR cuts as a function of the significance threshold, $Z^T$.

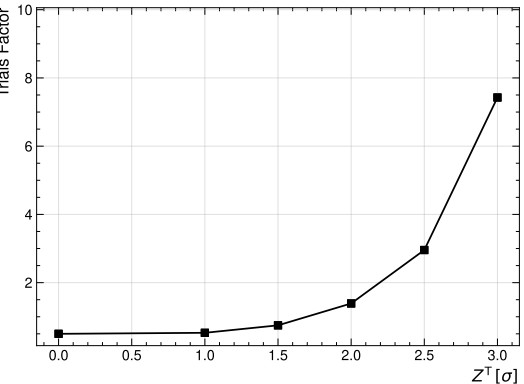

(b) The ratio of global p-value to local p-value, representing the trials factor, as a function of the significance threshold, $Z^T$.

Figure 6: Quantification of introduced look-elsewhere effect using BR ratios and the trials factor.

Figure 6b shows the trials factor as a function of the significance threshold $Z^T$. For lower significances, up to approximately $1.5\sigma$, the trials factor is between 0.5 and 1, indicating minimal inflation in the observed significance. However, at higher significances above $2\sigma$, the trials factor increases, reflecting the cumulative effect of multiple tests and the influence of the semi-supervised classifier.

These results demonstrate that the additional look-elsewhere effect introduced by the semi-supervised classifier is manageable and that the semi-supervised techniques employed here effectively maintain additional look-elsewhere effects within acceptable bounds. This suggests that the classifier's influence does not significantly inflate the observed significance, thereby preserving the integrity of the search for a narrow resonance at 150 GeV.

## 5 Conclusion

In this article we critically evaluated the efficacy of semi-supervised NN classifiers within the realm of narrow resonance searches in high-energy physics, gauging their tendency to produce additional look-elsewhere effects in the form of fake signals related to the selection of the background rejection cut. For this, we first generated the $Z\gamma$ MC dataset introduced in Section 2.1, which is then used to train and optimise the WGAN, VAE+D and KDE generative models. The optimised models were compared, revealing their ability to generate high-quality HEP data based on the training set of real MC-generated data. The best-performing generative model, for the frequentist study, was found to be the KDE. The pre-trained KDE model was then selected to generate 125′000 new training datasets, of 80′000 events each. On each of these data sets, we trained the NN and considered the outcome as a pseudo-experiment for a frequentist study of the look-elsewhere effect introduced by the semi-supervised NN.

The analysis reveals that while the look-elsewhere effect introduced by the semi-supervised classifier is minimal at lower significance values, it becomes more pronounced as the p-value decreases, with the trials factor reaching a maximum of 7.5 at the $3\sigma$ threshold (see Figure 6b). However, the trials factor, alongside the ratio of p-values for different BR cuts relative to the baseline 0% BR cut, indicates that the overall impact of this effect remains constrained and well-managed. These findings underscore the effectiveness of the semi-supervised NN classifier in maintaining control over fake signals, even as the background rejection becomes more stringent.

Importantly, this study demonstrates that the classifier's ability to discriminate between events is robust, with only a controlled introduction of the look-elsewhere effect. This confirms the viability of semi-supervised NN classifiers in HEP analyses, suggesting their potential to contribute meaningfully to future discoveries. The methodology employed here, using the $Z\gamma$ background process as a baseline, sets the stage for more comprehensive investigations. Future research should build on this foundation by incorporating more traditional physics scenarios and exploring the inclusion of signal processes to further validate and refine the approach.

# Acknowledgments

We are thankful to Rachid Mazini for his constructive discussions and suggestions regarding the analysis.

**Funding information** Support of the University of the Witwatersrand as well as the South African Department of Science and Innovation through the SA-CERN program is gratefully acknowledged. The work of A.C. is supported by a professorship grant from the Swiss National Science Foundation (No. PP00P21_76884).

# A   Example plots of fits to signal and background

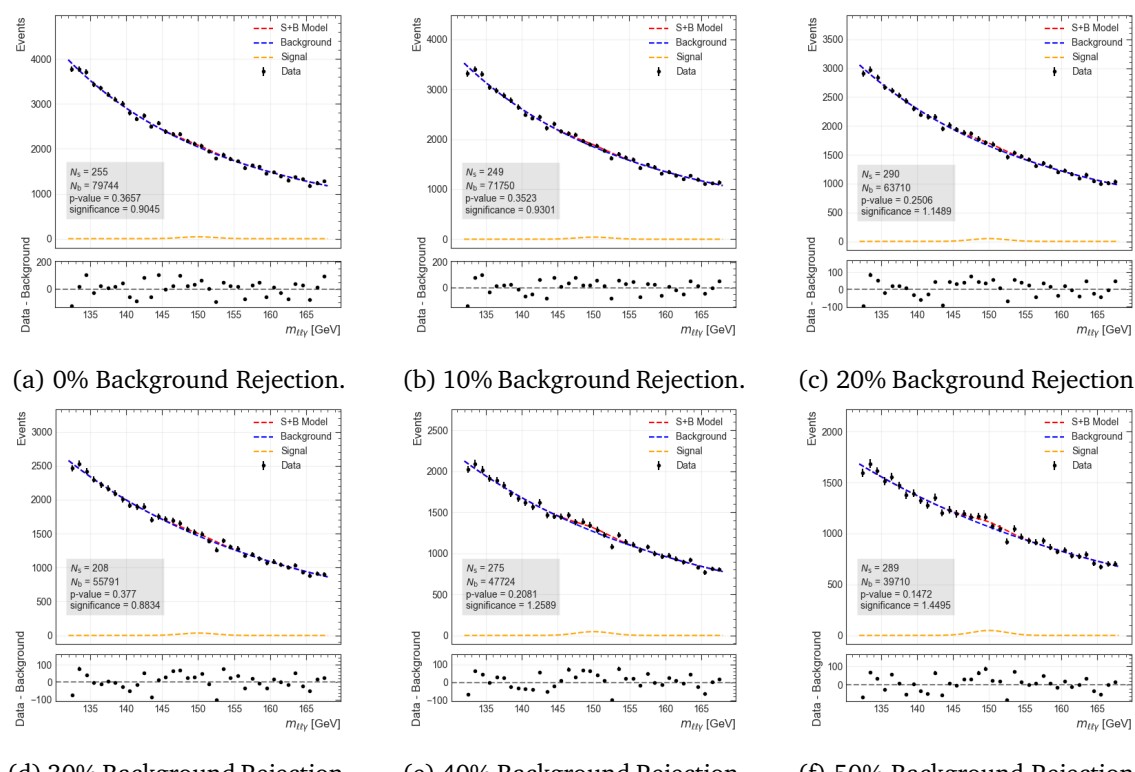

Figure 7: Signal and background fits for an example pseudo-experiment. Note that we only allow for a resonance within the signal region whose peak is centered at 150 GeV.

In each of the frequentist study's pseudo-experiments, the six background rejection samples are each used to quantify the extent of fake signals in the mass window of the $m_{\ell\ell\gamma}$ distribution. To achieve this, the $m_{\ell\ell\gamma}$ distributions are fit with the signal and background function. Figure 7 presents the $m_{\ell\ell\gamma}$ distribution of the six different background rejection samples, including the signal and background fits for a selected pseudo-experiment.

# B    Machine learning data generation

In this section, we introduce the ML-based data generation methods compared within this study. The KDE, WGAN and VAE+D methods are each introduced and their architecture and hyper-parameters are presented. This is followed by a visual demonstration of the ability of each model in the form of generated feature distributions and event-wise feature correlation.

## B.1    Kernel density estimator (KDE)

Kernel density estimation (KDE) is a non-parametric method for estimating the probability density function of a given random variable. The method is an unsupervised machine learning technique used to generate large samples of events that accurately reflect the statistics of real physics [65–67]. The KDE is implemented using the Scikit-learn library [68]. Through a scan of bandwidth parameters, the optimal value for this analysis was determined to be $1 \cdot 10^{-3}$. The resultant feature distributions and correlation plots are shown in Figures 8 and 9, respectively.

## B.2    Wasserstein generative adversarial network (WGAN)

The Generative Adversarial Network (GAN) training strategy is an interaction between two competing neural networks. The generator model, $G$, maps a source of noise to the desired feature space. A discriminator network receives both a generated sample and a MC data sample and is trained to distinguish between the two. The generator is trained to output realistic data, while the discriminator is simultaneously trained to distinguish the generated data from the MC data. An improved methodology of the GAN, described in Ref. [69], is the Wasserstein GAN (WGAN). This model adopts the Wasserstein distance, $W(q, p)$, which is defined as the minimum cost required to transform the probability distribution $q$ into the distribution $p$ by moving probability mass efficiently. The discriminator is replaced in the improved model with a critic, $C$, and the gradients are controlled using a gradient penalty, $GP$. The $GP$ penalises the normal of the critic gradients with respect to the input. The optimisation of the WGAN is a delicate interaction between the architecture and hyper-parameters of the models. The model is optimised on various combinations of architectures and hyper-parameters. The final optimised model consists of the following architectures and hyper-parameters. The Critic network consists of an input layer, three hidden layers and an output layer. The input layer has 18 nodes corresponding to the input features, the hidden layers have 256, 512 and 256 nodes, respectively, and a single neuron in the output layer. The critics neurons are activated using the ReLu activation function. The generator network consists of an input layer of 16 nodes representing latent space, four hidden layers of 256, 512, 1024, and 256 nodes respectively. The output layer consists of 18 neurons representing the input features. The generator network uses a combination of batch normalisation and ReLu as activation functions. The optimised hyper-parameters used for both the generator and critic are a learning rate, batch size, and gradient penalty constant, $\lambda$, of $1 \cdot 10^{-4}$, 256, and $1 \cdot 10^{-3}$ respectively. To improve the ability of the critic with respect to the generator, the critic is trained five times for each time the generator is trained. The resultant generated feature distributions and correlation plots are shown in Figures 10 and 11 respectively.

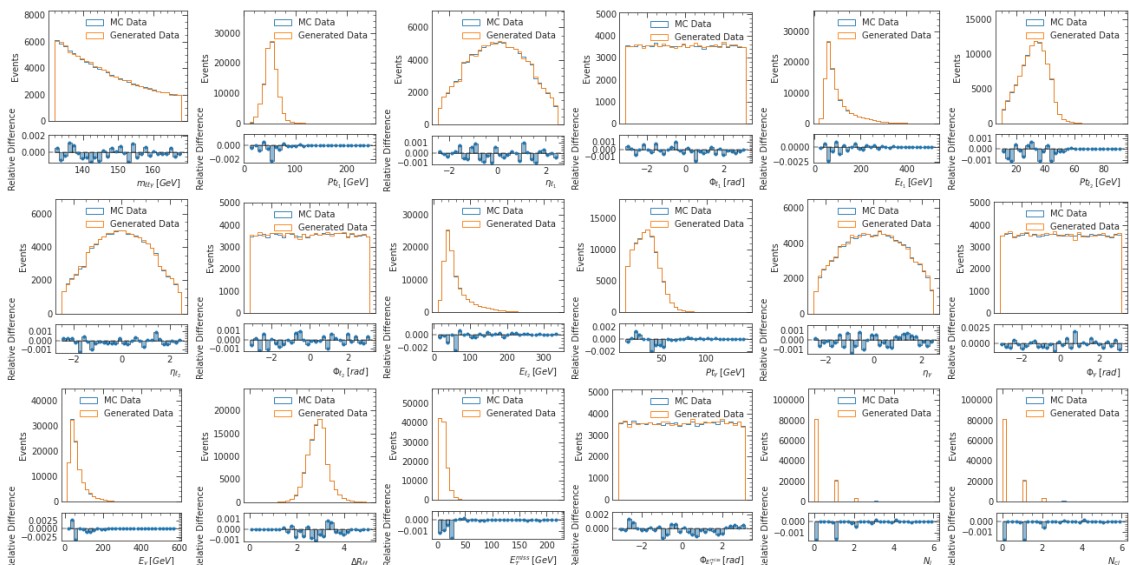

Figure 8: Feature distribution comparison between the KDE generated events and the MC events. The figures demonstrate that KDE generated feature distributions accurately reflect those of the MC dataset.

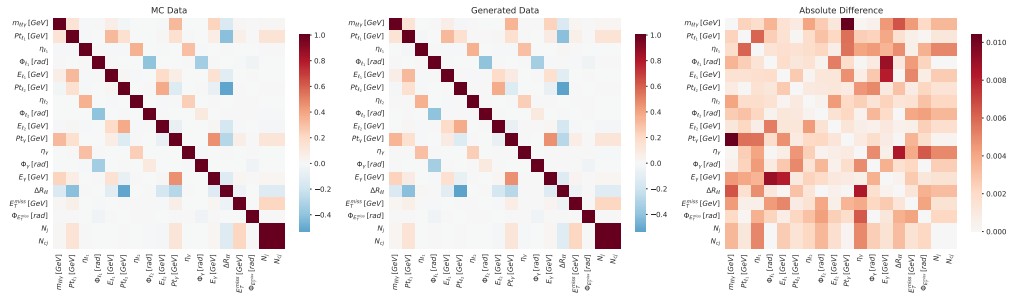

Figure 9: Feature correlation comparison between the MC and KDE generated events. Left: feature correlation of MC training dataset. Middle: feature correlation of KDE generated dataset. Right: the absolute difference between the feature correlations of the MC and KDE generated datasets. The feature correlation plots show that the maximum absolute difference between KDE generated events and MC events is 0.01.

### B.3 Variational auto-encoder + discriminator (VAE+D)

The Variational Autoencoder with Adversarial Training (VAE+D) is a hybrid data generation approach that combines Variational Autoencoders (VAEs) with adversarial training. VAEs are generative models used for learning latent data patterns. They encode data into a latent space and decode it back to the original space. However, VAEs can struggle with generating detailed and diverse samples. VAE+D addresses this limitation by incorporating a discriminator network, inspired by GANs. In VAE+D, the generator not only minimizes reconstruction errors but also aims to produce samples that are indistinguishable from real data according to the discriminator's judgment. The optimised VAE+D model includes the optimised encoder, decoder and discriminator networks. The encoder, decoder and discriminator networks each have three hidden layers of 256 nodes each. The latent space input to the encoder consists of 16 nodes and the output to the decoder consists of 18 neurons representing the input features.

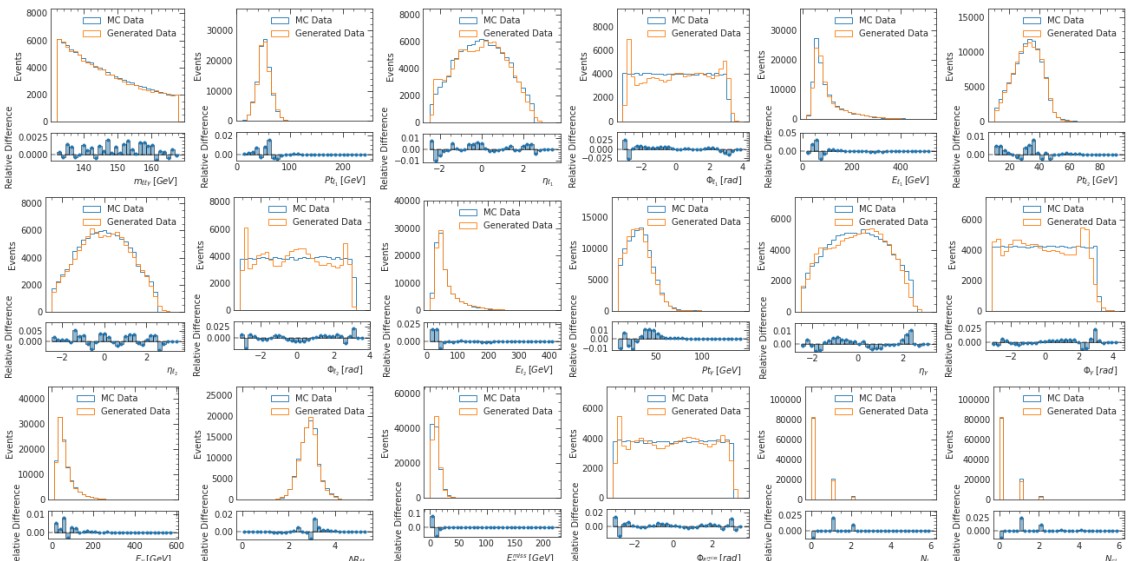

Figure 10: Feature distribution comparison between the WGAN generated events and the MC events. The figures demonstrate that the majority WGAN generated feature distributions accurately reflect those of the MC dataset.

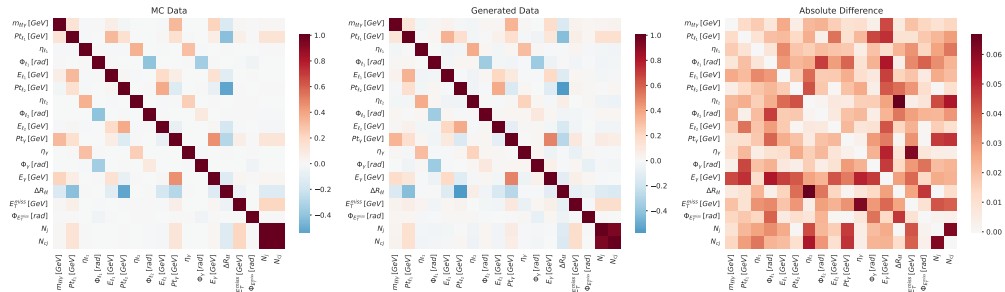

Figure 11: Feature correlation comparison between the MC and WGAN generated events. Left: feature correlation of MC training dataset. Middle: feature correlation of WGAN generated dataset. Right: the absolute difference between the feature correlations of the MC and WGAN-generated datasets. The feature correlation plots show that the maximum absolute difference between WGAN-generated events and MC events is 0.06.

The discriminator network has an input layer of 18 neurons reflecting the input features and a single output neuron. The eLu activation function is used to activate neurons in each network. A learning rate of $5 \cdot 10^{-4}$, batch size of 256 and scalar parameter $\gamma$ of 500 are used for training. The final optimised resultant feature distributions and correlations are presented in Figures 12 and 13, respectively.

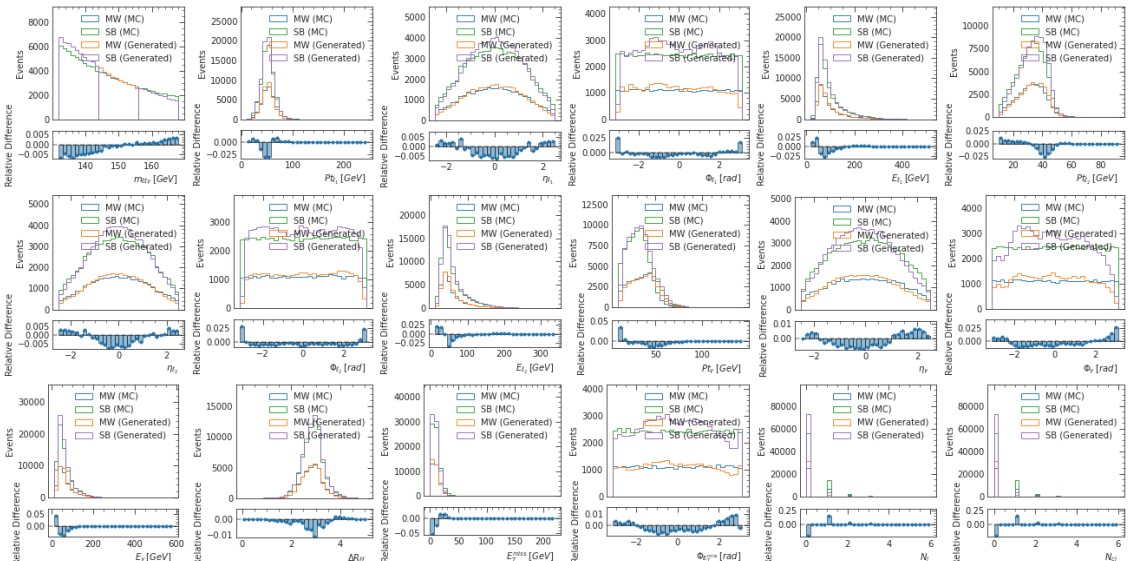

Figure 12: Feature distribution comparison between the VAE+D generated events and the MC events. Each distribution is shown in terms of the events in the mass-window (MW) and side band (SB). The figures demonstrate that the majority VAE+D generated feature distributions substantially reflect those of the MC dataset.

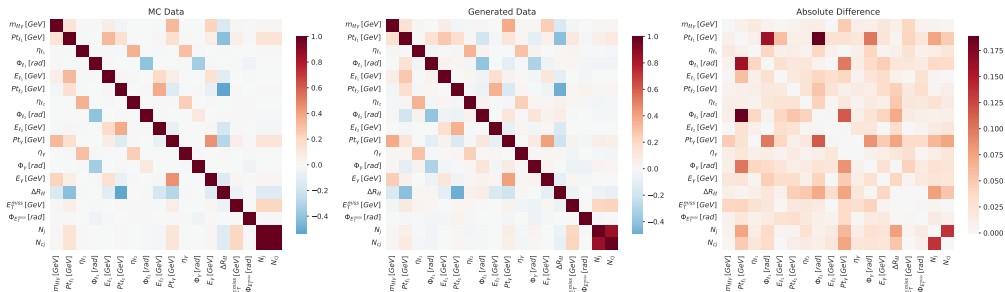

Figure 13: Feature correlation comparison between the MC and VAE+D generated events. Left: feature correlation of MC training dataset. Middle: feature correlation of VAE+D generated dataset. Right: the absolute difference between the feature correlations of the MC and VAE+D generated datasets. The feature correlation plots show that the maximum absolute difference between VAE+D generated events and MC events is 0.18.

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
