# Peer review of "Trials Factor for Semi-Supervised NN Classifiers in Searches for Narrow Resonances at the LHC"

_SciPost Physics Core, doi:SciPost Phys. Core 7, 073 (2024)_

## Round 1 · Referee Report · Anonymous (Referee 1) · 2024-5-6

Report

The manuscript introduces an innovative approach for detecting anomalies indicative of the production and decay of a resonance beyond the Standard Model. Employing classifiers constructed within a semi-supervised neural network framework, the method ensures assessment of look-elsewhere effects. A demonstration of the approach is provided in the context of potentially observing a signal within the $Z\gamma$ final state.

While the subject matter holds significant interest and the manuscript could certainly warrant consideration for publication within SciPost Physics Core, enhancing its comprehensiveness through further clarifications and elaborations would be beneficial. Specifically, it would be advisable for the authors to explore one or two additional illustrations of their method, including for instance conventional resonance searches with final states comprising a pair of jets, leptons, or photons, and to assess the impact of background systematics.

I now proceed with a list of comments that should be addressed by the authors.

1 - The introduction of the article could benefit from a discussion on experimental new physics searches utilising unsupervised machine learning methods, as well as a more comprehensive explanation of the distinctions/similarities between the proposed semi-supervised technique and other (semi-supervised) methods used for anomaly detection. The conclusions should be rewritten to reflect these considerations.

Furthermore, clarification is needed regarding the assertion that the proposed method is less model-dependent than other methods, especially semi-supervised or unsupervised ones.

2- The manuscript illustrates its methodology through resonance searches in the $Z\gamma$ final state. While the choice of this example is motivated by existing anomalies in LHC data, care should be taken to streamline the referencing and ensure clarity and conciseness in the justification. Consideration should be given to replacing the second paragraph on page 3 with a succinct statement detailing the rationale behind choosing the $Z\gamma$ analysis as an illustrative example of the proposed methodology. The heavy reliance on 16 self-citations out of 21 references, which exclude many relevant experimental papers, is in my opinion unnecessary in light of the actual topic of the present manuscript.

Furthermore, it is essential to accurately characterise the origin of these anomalies. Properly distinguishing between those confirmed by LHC collaborations and those proposed by phenomenological works, which may lack access to comprehensive statistical treatment, is crucial.

In addition, as written above, including other illustrative examples based on standard resonance searches in dijet, diphoton or dilepton final states, would be beneficial for readers.

3 - Section 2.1 lacks sufficient information on the simulation toolchain used. Event generation for the $pp\to Z\gamma \to \ell\ell\gamma$ process seems to enforce the intermediate $Z$-boson to be on-shell. However, since the mass window in $m_{\ell\ell\gamma}$ is large enough, off-shell $Z$ contributions, virtual-photon contributions, and their interference are relevant. It remains unclear whether they have been properly accounted for.

Furthermore, the discussion on the chosen parton density set is unclear. It is essential to clarify whether next-to-leading-order matrix elements have been consistently convolved with next-leading-order parton densities, and not leading-order ones.

Finally, the text does not clearly distinguish between generator-level cuts and reconstructed-level cuts that are implemented in the simulation. Providing a clear delineation between these sets of cuts is crucial. Additionally, Section 2.1 should include details on preselection criteria, like cuts on the number of leptons and photons, that are currently not discussed.

4 - The manuscript should define central jets and specify the associated pseudo-rapidity cut.

5 - Figures 1 and 2 should be adjusted to improve readability. The missing transverse energy spectrum could be presented with a log scale or a reduced domain to enhance clarity. Additionally, in figure 2, all eight lower insets should indicate whether they refer to the sideband or signal mass window. In fact, consideration should be given to showing both these curves.

6 -The caption of figure 4 should define the acronym 'BR' for clarity.

7 - In Section 3.3, the manuscript should avoid using the term 'centre of mass' to refer to the 'center of the signal mass window', as 'centre of mass' has a different well-defined meaning.

8 - It would be instructive to assess the impact of background systematics on the calculation of local significance, especially considering the incomplete background modeling acknowledged by the authors in Section 2. Equation (3) should be generalised accordingly, and the results of Section 4 updated subsequently.

9 - The bibliography should be carefully proofread to correct any errors. Specifically, attention should be given to identifying and correcting duplicate references (like references [2] and [3]), updating references that are now published (like reference [23]), and ensuring insertion of complete references (like [45] and [46]).

Recommendation

Ask for major revision

---

## Round 1 · Referee Report · Anonymous (Referee 2) · 2024-5-24

Strengths

1- The paper highlights and addresses an important issue namely an additional multiple-testing problem (aka “the look elsewhere effect”) that arises when weakly-supervised models are used in searches for new particles. 2- The investigation of different generative models (WGAN, VAE, KDE) to create fast simulators yielded useful information about the effectiveness of these models for the kind of data used. 3- The methodology presented is a step in the right direction in trying to quantify the additional look elsewhere effect. 4- The steps in the methodology are clear but see Weaknesses.

Weaknesses

1- While the methodology is a step in the right direction, it is far from clear that the proposed way to handle the statistically dependent Z-values or, equivalently the statistically dependent p-values, is superior to methods that are already in use for the standard look elsewhere effect. 2- The authors point out correctly that the NN classifier needs to be statistically independent of the lepton-lepton-photon mass because the latter is used to define the two samples that are used for training, but they do not provide evidence that this is the case. 3- There is some inconsistency in notation in Eqs. (1), (2) and (4). The symbols for invariant mass and Z seem to change. The invariant mass is denoted by a symbol with lepton and photon subscripts in the text but is denoted by m in Eq. (2). In Eq. (3) the symbol Z is used for signal significance. But in Section 4, which introduces the “probability density functions (PDF) of the significances”, the presumption is that these are the PDFs of the Z-values. If so, one might have expected to see symbols such as fi (Zi), i = 1,…,6 for each of the six significances Zi, but instead one sees the symbol f (sigma). Since Z (from Eq. (3), which is a well-known generalization of Ns / sqrt(Nb)) can be interpreted as the “number of standard deviations above the background”, it is unclear what “sigma” is intended to represent in f (sigma). It could be sigma = Z sqrt(Nb)), but that does not seem to be what is intended. This needs to be clarified. Furthermore, from Eq. (3) it is unclear why the domain of f (sigma) should extend to negative values. That also needs clarification. 4- It is unclear why the fits are not performed so that they give Ns and Nb directly. For example, if the standard particle physics toolkit RooFit were used that would be the case since the probability densities are automatically normalized. 5- In Section 4, it is unclear why the average PDF (Eq. (6)) is the appropriate quantity to use to compute the global p-value. (See discussion in Report.)

Report

Since the purpose of using a weakly supervised NN is to look for statistically significant signals without bias, the authors need to strengthen the motivation for their specific analysis of the pseudo-experiments. For example, one might argue that one should take the largest Z-value under the null hypothesis and use its PDF to assess how frequently a >= Z-sigma fake signal occurs. Ultimately, the trials factor is not what is of interest, but rather getting the p-value that automatically incorporates that factor.

In Section 4, the authors note correctly that the six Z-values are not statistically independent, but then argue in the next sentence that “This approach facilitates an examination of the impact of background rejection on the results…”. This reviewer would agree if the goal were to answer the question: “Given this level of background rejection what is the probability to obtain a signal significance Z (as defined in Eq. (3)) equal to or larger than a specified number”. But to quantify the look-elsewhere effect induced by the NN it is necessary, somehow, to deal with the six Z-values. The authors’ proposal in Eq. (6) suffers from the lack of clarity about the intended meaning of “sigma”. But if sigma really means Z, it is unclear why the average PDF, Eq. (6), which is billed as the “global” PDF is the appropriate quantity to use to compute the global p-value. Is it not necessary to account for the statistical dependencies between the Z-values if all six are used?

Requested changes

1- Remove inconsistencies in notation. 2- Motivate the use of Eq. (6). 3- Given an analysis that yields K statistically dependent p-values (or Z-values) explain why methods such as the Bonferroni correction are not sufficient to account for this particular look elsewhere effect or why methods already in use in particle physics cannot be applied to this look elsewhere effect.

Recommendation

Ask for major revision

---

## Round 2 · Referee Report · Anonymous (Referee 1) · 2024-7-6

Report

I would like to thank the authors for following my suggestions and addressing my comments. However, there are still two items (my previous comments #2 and #3) that have not been satisfactorily considered.

1 - Former comment #2. I still find it exaggerated that nearly half of the bibliography of this paper (35 references out of 73) is necessary to justify the choice of the $Z\gamma$ example, that is just a showcase for the methodology. Moreover, among these 35 references, there are 15 self-citations (specifically, the entire set of chosen theory papers discussing the multi-lepton anomalies).

Additionally, the text is written in a way that might lead readers to believe that a model in which the Standard Model is extended by three scalars with well-defined masses has been confirmed by data. This is not the case, as such a confirmation could only be made by the LHC experiments (and not by any theoretical study). This should be made crystal clear, and the text should be rephrased accordingly.

Finally, the authors mentioned that they chose the multi-lepton anomalies as a showcase for the proposed method. While this is fine, I do not understand their reluctance to add a second example. As they stated in their reply to my report, the methodology is independent of the chosen physics case. Therefore, there should be no reason to refuse to add a second, different and maybe more traditional, physics example.

2 - Former comment #3. The description of the simulation tool chain used in this work is now very clear, and I thank the authors for adding the associated text in section II.1. However, I do not understand why virtual-photon contributions have been ignored in the simulation of the signal. The process considered is $pp \to \ell\ell\gamma$ with the lepton pair invariant mass being off the $Z$ peak. Therefore, there is no reason to only consider $Z$-mediated diagrams, as the impact of the virtual-photon component on the signal is not negligible. This should be trivial to fix with MG5aMC.

Furthermore, the authors should explicitly state which PDF set they have used. Is it NNPDF 2.3 as suggested by the reference quoted?

Recommendation

Ask for minor revision

---

## Round 2 · Referee Report · Anonymous (Referee 2) · 2024-8-8

Strengths

  1. The authors have addressed my comments and suggestions.

Weaknesses

  1. I still have quibbles, but there are no showstoppers.

Report

The paper passes the bar.

Requested changes

None.

Recommendation

Publish (meets expectations and criteria for this Journal)

---

## Round 2 · Referee Report · Anonymous (Referee 3) · 2024-8-11

Report

This paper is about the trials factor in a variant of weakly supervised anomaly detection at the LHC, which as far as I can tell is just the usual CWoLa method but with multiple thresholds on the anomaly score. The authors are interested in quantifying a trials factor associated with performing multiple simultaneous bump hunts with these different thresholds.

I have one major concern about the methodology of this paper. To the best of my understanding, the trials factor being estimated here is not the usual trials factor considered in HEP. The usual trials factor would involve comparing the nominal value Zmax=max(Z1,...,Z6) -- the max local significance across the different thresholds (six in this paper) -- and the true significance obtained by constructing the distribution of Zmax using pseudoexperiments.

Instead, the authors are considering a different kind of trials factor. They are comparing (a) the p-value p(Z) corresponding to a Z score, calculated from the CDF of the distribution obtained from flattening all 6 Z scores across all pseudoexperiments into a single distribution; to (b) the p(Z) derived from the CDF of the Z-score distribution for the inclusive bump hunt.

They are referring to (a) as the global significance and (b) as the local significance. I think I'm okay with calling (a) a global significance, but I don't understand why it should be compared only against the inclusive Z-score. Shouldn't it also be compared against the Z-scores from all the other CWoLa thresholds as well?

The authors should also acknowledge/clarify that the trials factor they are considering here differs from the other type of trials factor often considered in HEP.

Some more specific points of feedback are:

1) References 19-26 are woefully incomplete. The authors should do a more thorough and proper job surveying the literature and add many more references on the applications of weak supervision to anomaly searches at the LHC. A lot of work has been done by many authors on this topic and the authors are not giving the proper credit where credit is due.

2) "therefore the training samples should be indistinguishable to the NN apart from statistical fluctuations."

What if the SB and SR events have systematic differences (ie the features are not uncorrelated with m)? The authors do not demonstrate that the CWoLa method is even valid here.

3) I don't understand Fig 5b, why is the p-value more significant than expected from Z_T for lower background rejection fractions (eg for 0% selection, the p-value is 3 sigma when Z_T is 2 sigma)?

4) If the NN sculpts the m distribution this will also inflate p-values. How can the authors be sure that it is a LEE and not sculpting?

Recommendation

Ask for major revision

---

## Round 2 · Author Response

Dear Editors,

We sincerely thank the referees for their thorough and thoughtful review of our paper. Their insights are invaluable to us, and we have addressed each of their comments in detail in the following response.

Kind Regards,
Benjamin Lieberman and authors

---

## Round 2 · List of Changes

Response to Referee report 1 Comments:
Comment 1: The introduction of the article could benefit from a discussion on experimental new physics searches utilising unsupervised machine learning methods, as well as a more comprehensive explanation of the distinctions/similarities between the proposed semi-supervised technique and other (semi-supervised) methods used for anomaly detection. The conclusions should be rewritten to reflect these considerations.
Furthermore, clarification is needed regarding the assertion that the proposed method is less model-dependent than other methods, especially semi-supervised or unsupervised ones.
Response 1:
We have improved the introductory paragraphs introducing the proposed semi-supervised technique. This includes the introduction and comparison of unsupervised, semi-supervised and weakly supervised methods with corresponding added references. Furthermore, we have added a discussion comparing the extent of model dependence. Although clarification that the proposed method is less model-dependent than alternative semi- or weakly supervised would provide a valuable study, it is largely dependent on the specific signal, background and region of interest and is not the focus of this study.

Comment 2: The manuscript illustrates its methodology through resonance searches in the 𝑍𝛾 final state. While the choice of this example is motivated by existing anomalies in LHC data, care should be taken to streamline the referencing and ensure clarity and conciseness in the justification. Consideration should be given to replacing the second paragraph on page 3 with a succinct statement detailing the rationale behind choosing the 𝑍𝛾 analysis as an illustrative example of the proposed methodology. The heavy reliance on 16 self-citations out of 21 references, which exclude many relevant experimental papers, is in my opinion unnecessary in light of the actual topic of the present manuscript.
Furthermore, it is essential to accurately characterise the origin of these anomalies. Properly distinguishing between those confirmed by LHC collaborations and those proposed by phenomenological works, which may lack access to comprehensive statistical treatment, is crucial.
In addition, as written above, including other illustrative examples based on standard resonance searches in dijet, diphoton or dilepton final states, would be beneficial for readers.

Response 2: In our analysis we selected the 𝑍𝛾 final state, motivated by the multi-lepton anomalies as an ideal showcase for an analysis of semi-supervised learning for narrow resonance with topological requirements. Although 𝛾𝛾 is similarly motivated by the multi-lepton anomalies, we selected 𝑍𝛾. The methodology and results stand independent of the anomaly at 152GeV and use it only as a showcase. Therefore even if 152GeV excess goes away, the paper will still stand as a showcase. We have removed unnecessary self-citations and added relevant references from the LHC collaborations to better substantiate our motivation.

Comment 3: Section 2.1 lacks sufficient information on the simulation toolchain used. Event generation for the 𝑝𝑝→𝑍𝛾→ℓℓ𝛾 process seems to enforce the intermediate 𝑍-boson to be on-shell. However, since the mass window in 𝑚ℓℓ𝛾 is large enough, off-shell 𝑍 contributions, virtual-photon contributions, and their interference are relevant. It remains unclear whether they have been properly accounted for.
Furthermore, the discussion on the chosen parton density set is unclear. It is essential to clarify whether next-to-leading-order matrix elements have been consistently convolved with next-leading-order parton densities, and not leading-order ones.
Finally, the text does not clearly distinguish between generator-level cuts and reconstructed-level cuts that are implemented in the simulation. Providing a clear delineation between these sets of cuts is crucial. Additionally, Section 2.1 should include details on preselection criteria, like cuts on the number of leptons and photons, that are currently not discussed.

Response 3:
Our simulation accounts for the fact that the ℓℓγ invariant mass of 130 to 170 GeV necessitates the Z boson to be off-shell. We configured MadGraph to include off-shell Z boson contributions, although a prompt photon is considered. The event generation utilized next-to-leading-order (NLO) parton distribution functions (PDFs) convolved with NLO matrix elements, ensuring consistency and accuracy in the simulation. Generator-level cuts were applied during the event generation in MadGraph, specifically imposing an invariant mass cut on the ℓℓγ system to select events within the 130-170 GeV range. These generated events were then processed through PYTHIA for parton showering and DELPHES for detector simulation, where reconstructed-level cuts were applied to mimic experimental conditions. Reconstructed-level cuts included detailed preselection criteria such as the number of leptons and photons, their transverse momenta, and isolation requirements. Furthermore, overlap removal procedures were implemented to avoid double-counting objects. This involved removing jets that were too close to leptons or photons to ensure distinct identification of each particle. These updates, along with more detailed descriptions, have been included in the revised version of the paper.

Comment 4: The manuscript should define central jets and specify the associated pseudo-rapidity cut.

Response 4: We have added the definition and associated pseudo-rapidity cut for central jets to Section 2.1. Monte Carlo Simulation.

Comment 5: Figures 1 and 2 should be adjusted to improve readability. The missing transverse energy spectrum could be presented with a log scale or a reduced domain to enhance clarity. Additionally, in figure 2, all eight lower insets should indicate whether they refer to the sideband or signal mass window. In fact, consideration should be given to showing both these curves.

Response 5: We have updated Figures 1 and 2 to improve readability. Firstly the domain of the missing transverse energy spectrum, the number of jets and the number of central jets have been reduced. Secondly the overlap between the plots and the legends has been removed. Finally, in Figure 2 the lower insets have been updated to include the relative difference for both the side band and mass window categories with corresponding legends.

Comment 6: The caption of figure 4 should define the acronym 'BR' for clarity.

Response 6: We have added a definition for the acronym ‘BR’ in the caption of Figure 4 for improved clarity.

Comment 7: In Section 3.3, the manuscript should avoid using the term 'centre of mass' to refer to the 'center of the signal mass window', as 'centre of mass' has a different well-defined meaning.

Response 7: To avoid misinterpretation we have replaced our use of ‘center of mass’ with ‘center of the signal mass window’ as suggested.

Comment 8: It would be instructive to assess the impact of background systematics on the calculation of local significance, especially considering the incomplete background modeling acknowledged by the authors in Section 2. Equation (3) should be generalised accordingly, and the results of Section 4 updated subsequently.

Response 8: The study focuses on measuring the look elsewhere effect. As background systematics are additionally applied and conventionally applied elsewhere it has been factored out of this study and can be applied after. The background systematics must therefore be applied additionally to this case study before it can be applied to a specific excess. The background systematics are obtained by using different fitting functions which however does not impact the DNN part of the analysis.

To make this clear to the reader we have added a footnote (“Note that the systematics related to the background fitting functions, i.e. the spurious signal analysis, are not included here. However, this uncertainty should be included on top of the additional look-elsewhere effect studied here and is not related to the use of NNs.”) following Equation 3.

Comment 9: The bibliography should be carefully proofread to correct any errors. Specifically, attention should be given to identifying and correcting duplicate references (like references [2] and [3]), updating references that are now published (like reference [23]), and ensuring insertion of complete references (like [45] and [46]).

Response 9: Firstly, we apologize for the extent of error in the bibliography. We have removed duplicate references, updated references to now published articles, and corrected incomplete references.

Response to Referee report 2 Comments:
Weaknesses:
Comment (weakness) 1: While the methodology is a step in the right direction, it is far from clear that the proposed way to handle the statistically dependent Z-values or, equivalently the statistically dependent p-values, is superior to methods that are already in use for the standard look elsewhere effect.

Response (weakness) 1: The methodology is designed to evaluate the potential look-elsewhere effect arising from semi-supervised classifiers, rather than the standard look-elsewhere effect. This method employs the frequentist framework to calculate this effect directly (through statistics of multiple tests), instead of relying on the approximations typically used for calculating the classic look-elsewhere effect.

Comment (weakness) 2: The authors point out correctly that the NN classifier needs to be statistically independent of the lepton-lepton-photon mass because the latter is used to define the two samples that are used for training, but they do not provide evidence that this is the case.

Response (weakness) 2: To make clear the dependence of the NN training samples on the lepton-lepton-photon mass, we have added the definitions of the side band and mass window categories to Section 2.1. (previously only in Section 3.3).

Comment (weakness) 3: There is some inconsistency in the notation in Eqs. (1), (2) and (4). The symbols for invariant mass and Z seem to change. The invariant mass is denoted by a symbol with lepton and photon subscripts in the text but is denoted by m in Eq. (2). In Eq. (3) the symbol Z is used for signal significance. But in Section 4, which introduces the “probability density functions (PDF) of the significances”, the presumption is that these are the PDFs of the Z-values. If so, one might have expected to see symbols such as fi (Zi), i = 1,…,6 for each of the six significances Zi, but instead one sees the symbol f (sigma). Since Z (from Eq. (3), which is a well-known generalization of Ns / sqrt(Nb)) can be interpreted as the “number of standard deviations above the background”, it is unclear what “sigma” is intended to represent in f (sigma). It could be sigma = Z sqrt(Nb)), but that does not seem to be what is intended. This needs to be clarified. Furthermore, from Eq. (3) it is unclear why the domain of f (sigma) should extend to negative values. That also needs clarification.

Response (weakness) 3: With regards to notation please see requested changes 1 and response.
In Equation 3 the domain is extended to negative values to allow a full analysis of the dynamics of each sample. If a negative signal yield is found then the corresponding significance must reflect that. This is used to verify the fitting process and therefore confirm the validity of the positive only values of interest.

Comment (weakness) 4: It is unclear why the fits are not performed so that they give Ns and Nb directly. For example, if the standard particle physics toolkit RooFit were used that would be the case since the probability densities are automatically normalized.

Response (weakness) 4: The fit methodology was initially achieved using the RooFit toolkit as well as asymptotic calculator to calculate the significance as suggested. This was repeated using the more “manual” process described in the paper (after verifying the results reflected those of the internal toolkit methods) to provide a more comprehensive and clearer methodology.

Comment (weakness) 5: In Section 4, it is unclear why the average PDF (Eq. (6)) is the appropriate quantity to use to compute the global p-value. (See discussion in Report.)

Response (weakness) 5: Please see requested changes 2 and response.
Requested Changes:
Comment (requested change) 1: Remove inconsistencies in the notation

Response (changes) 1:
To improve consistency of notation we have updated equations 2 to use the correct mass notation (mℓℓγ). We have updated Equations 4, 5, 6 and 7 as well as Figure labels and in text references to use “Z” for significance rather than misleading use of σ (e.g. changed f(σ)BR to f(Z)BR).

Comment (requested change) 2: Motivate use of Eq. 6 (... it is unclear why the average PDF, Eq. (6), which is billed as the “global” PDF is the appropriate quantity to use to compute the global p-value. Is it not necessary to account for the statistical dependencies between the Z-values if all six are used?)

Response (changes) 2: We acknowledge that Eq. 6 was misleading. We have therefore removed Eq. 6 and replaced it with explanatory text. The global distribution of results is directly calculated as the ensemble or all the outcomes across all background rejections and not the average or weighted average. Although there are different numbers of events entering the fits of each BR, the number of fits/significance values for each BR is equal to the number of pseudo-experiments. Thus the global values include outputs from all BR to understand the dynamics of the semi-supervised response across all BRs without focusing on potential bias in individual categories (which is exposed in the comparative plots of local/BR distributions).

Comment (requested change) 3: Given an analysis that yields K statistically dependent p-values (or Z-values) explain why methods such as the Bonferroni correction are not sufficient to account for this particular look elsewhere effect or why methods already in use in particle physics cannot be applied to this look elsewhere effect.

Response (changes) 3: Methods such as Bonferroni and Vitells are approximations used to estimate the look-elsewhere effect (LEE). The frequentist framework used is a way of calculating the LEE directly providing a true depiction of the extent of induced error. Additionally, when calculating the look elsewhere effect generated from semi-supervised NN, one cannot assume that standard estimations are sufficient and must be calculated directly.

Once again, we extend our gratitude to the referee for their detailed and insightful comments. We trust that the comprehensive responses and revisions provided adequately address all the points raised. We are optimistic that our revised manuscript now meets the criteria for publication in SciPost.

---

## Round 4 · Referee Report · Anonymous (Referee 1) · 2024-9-9

Report

It seems that the only item left is an item on which the authors and I agree to disagree. I would have liked to see a second, more traditional, example showcasing the methodology. Whilst the authors agreed with me, they mentioned that this would represent too much work and won't bring much relatively to the current example.

I therefore leave the decision to the editor (and won't block the acceptance process).

Recommendation

Publish (meets expectations and criteria for this Journal)

---

## Round 4 · Referee Report · Anonymous (Referee 2) · 2024-10-4

Strengths

The current version of the paper is much clearer.

Weaknesses

On the whole, the weaknesses identified in previous versions of the paper have been adequately addressed.

Report

With he changes made, the paper is acceptable. I recommend publication.

Recommendation

Publish (meets expectations and criteria for this Journal)

---

## Round 4 · List of Changes

Referee 1: Comments and Response:

Dear Referee, we thank you for your continued review of our paper and constructive insight. We have carefully addressed each comment and provided responses below.

Comment 1.1. - Former comment #2. I still find it exaggerated that nearly half of the bibliography of this paper (35 references out of 73) is necessary to justify the choice of the Zγ example, that is just a showcase for the methodology. Moreover, among these 35 references, there are 15 self-citations (specifically, the entire set of chosen theory papers discussing the multi-lepton anomalies).
Additionally, the text is written in a way that might lead readers to believe that a model in which the Standard Model is extended by three scalars with well-defined masses has been confirmed by data. This is not the case, as such a confirmation could only be made by the LHC experiments (and not by any theoretical study). This should be made crystal clear, and the text should be rephrased accordingly.

Response 1.1. We thank the referee for his feedback and have re-written the introductory paragraph in question. We have improved the description of the model and anomalies used to motivate the selection of the Zγ dataset and reduced the number of self citations significantly. We believe that with these changes we provide a clear background and motivation to the selection of our showcase example without misleading the readers or using unnecessary self-citations.

Comment 1.2. - Finally, the authors mentioned that they chose the multi-lepton anomalies as a showcase for the proposed method. While this is fine, I do not understand their reluctance to add a second example. As they stated in their reply to my report, the methodology is independent of the chosen physics case. Therefore, there should be no reason to refuse to add a second, different and maybe more traditional, physics example.

Response 1.2. We agree with the referee that an additional example will be advantageous, however as performing the complete frequentist study takes hundreds of hours (±3 minutes per pseudo-experiment, with 125000 pseudo-experiment iterations), we believe the selected example is sufficient and can be used as a basis for further studies. However, we have added a sentence to the end of the conclusion stating that it would be beneficial to include in future studies.

Comment 2 - Former comment #3. The description of the simulation tool chain used in this work is now very clear, and I thank the authors for adding the associated text in section II.1. However, I do not understand why virtual-photon contributions have been ignored in the simulation of the signal. The process considered is pp→ℓℓγ with the lepton pair invariant mass being off the Z peak. Therefore, there is no reason to only consider Z-mediated diagrams, as the impact of the virtual-photon component on the signal is not negligible. This should be trivial to fix with MG5aMC.
Furthermore, the authors should explicitly state which PDF set they have used. Is it NNPDF 2.3 as suggested by the reference quoted?

Response 2. We apologize for not picking up this error which was due to an internal miscommunication. The virtual photon mediated diagram is considered and the statement in the paper has been amended. We thank the referee for this input as it will be important for future studies that include signal processes.
We thank the referee and have added the PDF set used, NNPDF 3.0, as suggested.

Referee 3: Comments and Response:

We thank the referee for their thorough review and valuable comments. After extensive internal discussions, we have agreed that the results should include the classical trials factor in addition to the version previously presented. Accordingly, we have re-written the results section to incorporate the calculation of the classical trials factor, which compares the probabilities of the maximum significances across all BR cuts and pseudo-experiments to the nominal values. This ensures a more comprehensive analysis and a clearer representation of the look-elsewhere effect in our study.

Comment 1: References 19-26 are woefully incomplete. The authors should do a more thorough and proper job surveying the literature and add many more references on the applications of weak supervision to anomaly searches at the LHC. A lot of work has been done by many authors on this topic and the authors are not giving the proper credit where credit is due.
Response 1: Thank you for your valuable feedback regarding the references. We have carefully reviewed your comment and have significantly expanded the literature citations in our manuscript, particularly in the area of weak supervision applied to anomaly searches at the LHC. We have added several key references to properly acknowledge the substantial body of work in this field and to give due credit to the contributions of many researchers. We believe this revision now more accurately reflects the extensive research efforts in this area.
Comment 2: "therefore the training samples should be
indistinguishable to the NN apart from statistical fluctuations."
What if the SB and SR events have systematic differences (ie the features are not uncorrelated with m)? The authors do not demonstrate that the CWoLa method is even valid here.
Response 2: Thank you for your comment. The features used to train the NN are carefully selected to not have correlation with the mass (mℓℓγ) as stated in section 2.1. “... The mℓℓγ distribution, and any features correlated with it, cannot be used to train the NN due to the fact that the training samples, mass window (144 < mℓℓγ < 156 GeV) and side-band (132 < mℓℓγ < 168 GeV excluding mass window), are defined on this mass.”). This was verified through the response of the neural network which consistently (for each pseudo-experiment) had AUC scores of ±0.5, providing evidence that no features have significant enough correlation with the mass to influence the NN.
Comment 3: I don't understand Fig 5b, why is the p-value more significant than expected from Z_T for lower background rejection fractions (eg for 0% selection, the p-value is 3 sigma when Z_T is 2 sigma)?
Response 3: In Figure 5b the p-values for lower background rejection fractions are more significant than expected from Z_T due to fewer positive significances yielded across all thresholds (as reflected in the distributions in Figure 5a). When extracting the corresponding p-values for positive significance thresholds this leads to higher significance from p-values which is exposed in Figure 5b. To make this clear we have added the following sentence to the results: “Note that, in Figure 5b, categories with lower background rejection differ more greatly

Comment 4: If the NN sculpts the m distribution this will also inflate p-values. How can the authors be sure that it is a LEE and not sculpting?
Response 4: Thank you for this insight. As we understand, sculpting occurs when the region of interest is close to the kinematic thresholds. We have therefore selected the mass range of 150GeV as it is far enough from kinematic limits (±100GeV) to make sure we are considering the LEE and not sculpting.

---

## Editorial Decision

published